# IALE: Imitating Active Learner Ensembles

## Abstract

Active learning (AL) prioritizes the labeling of the most informative data samples. However, the performance of AL heuristics depends on the structure of the underlying classifier model and the data. We propose an imitation learning scheme that imitates the selection of the best expert heuristic at each stage of the AL cycle in a batch-mode pool-based setting. We use DAGGER to train the policy on a dataset and later apply it to datasets from similar domains. With multiple AL heuristics as experts, the policy is able to reflect the choices of the best AL heuristics given the current state of the AL process. Our experiment on well-known datasets show that we both outperform state of the art imitation learners and heuristics.

## 1 Introduction

The high performance of deep learning on various tasks from computer vision (Voulodimos et al., 2018) to natural language processing (NLP) (Barrault et al., 2019) also comes with disadvantages. One of their main drawbacks is the large amount of labeled training data they require. Obtaining such data is expensive and time-consuming and often requires domain expertise.

Active Learning (AL) is an iterative process where during every iteration an oracle (e.g. a human) is asked to label the most informative unlabeled data sample(s). In *pool-based* AL all data samples are available (while most of them are unlabeled). In *batch-mode* pool-based AL, we select unlabeled data samples from the pool in acquisition batches greater than 1. Batch-mode AL decreases the number of AL iterations required and makes it easier for an oracle to label the data samples (Settles, 2009). As a selection criteria we usually need to quantify how informative a label for a particular sample is. Well-known criteria include heuristics such as model uncertainty (Gal et al., 2017; Roth & Small, 2006; Wang & Shang, 2014; Ash et al., 2020), data diversity (Sener & Savarese, 2018), query-by-committee (Beluch et al., 2018), and expected model change (Settles et al., 2008). As ideally we label the most informative data samples at each iteration, the performance of a machine learning model trained on a labeled subset of the available data selected by an AL strategy is better than that of a model that is trained on a randomly sampled subset of the data.

Besides the above mentioned, in the recent past several other data-driven AL approaches emerged. Some are modelling the data distributions (Mahapatra et al., 2018; Sinha et al., 2019; Tonnaer, 2017; Hossain et al., 2018) as a pre-processing step, or similarly use metric-based meta-learning (Ravi & Larochelle, 2018; Contardo et al., 2017) as a clustering algorithm. Others focus on the heuristics and predict the best suitable one using a multi-armed bandits approach (Hsu & Lin, 2015). Recent approaches that use reinforcement learning (RL) directly learn strategies from data (Woodward & Finn, 2018; Bachman et al., 2017; Fang et al., 2017). Instead of pre-processing data or dealing with the selection of a suitable heuristic they aim to learn an optimal selection sequence on a given task.

However, these pure RL approaches not only require a huge amount of samples they also do not resort to existing knowledge, such as potentially available AL heuristics. Moreover, training the RL agents is usually very time-intensive as they are trained from scratch. Hence, imitation learning (IL) helps in settings where very few labeled training data and a potent algorithmic expert are available. IL aims to train, i.e., *clone*, a policy to transfer the expert to the related few data problem. While IL mitigates some of the previously mentioned issues of RL, current approaches are still limited with respect to their algorithmic expert and their acquisition size (including that of Liu et al. (2018)), i.e., some only pick one sample per iteration, and were so far only evaluated on NLP tasks.

We propose an batch-mode AL approach that enables larger acquisition sizes and that allows to make use of a more diverse set of experts from different heuristic families, i.e., uncertainty, diversity,

expected model-change, and query-by-committee. Our policy extends previous work (see Section 2) by learning at which stage of the AL cycle which of the available strategies performs best. We use Dataset Aggregation (DAGGER) to train a robust policy and apply it to other problems from similar domains (see Section 3). We show that we can (1) train a policy on image datasets such as `MNIST`, `Fashion-MNIST`, `Kuzushiji-MNIST`, and `CIFAR-10`, (2) transfer the policy between them, and (3) transfer the policy between different classifier architectures (see Section 4).

## 2 RELATED WORK

Next to the AL approaches for traditional ML models (Settles, 2009) also ones that are applicable to deep learning have been proposed (Gal et al., 2017; Sener & Savarese, 2018; Beluch et al., 2018; Settles et al., 2008; Ash et al., 2020). Below we discuss AL strategies that are trained on data.

**Generative Models.** Explicitly modeled data distributions capture the *informativeness* that can be used to select samples based on diversity. Sinha et al. (2019) propose a pool-based semi-supervised AL where a discriminator discriminates between labeled and unlabeled samples using the latent representations of a variational autoencoder. The representations are used to pick data points that are most diverse and representative (Tonnaer, 2017). Mirza & Osindero (2014) use a conditional generative adversarial network to generate samples with different characteristics from which the most informative are selected using the uncertainty measured by a Bayesian neural network (Kendall & Gal, 2017; Mahapatra et al., 2018). Such approaches are similar to ours (as they capture dataset properties) but instead we model the dataset implicitly and infer a selection heuristic via imitation.

**Metric Learning.** Metric learners such as Ravi & Larochelle (2018) use a set of statistics calculated from the clusters of un-/labeled samples in a Prototypical Network's (Snell et al., 2017) embedding space, or learn to rank (Li et al., 2020) large batches. Such statistics use distances (e.g. Euclidean distance) or are otherwise converted into class probabilities. Two MLPs predict either a quality or diversity query selection using backpropagation and the `REINFORCE` gradient (Mnih & Rezende, 2016). However, while they rely on statistics over the classifier's embedding and explicitly learn two strategies (quality and diversity) we use a richer state and are not constrained to specific strategies.

**Reinforcement Learning (RL).** The AL cycle can be modeled as a sequential decision making problem. Woodward & Finn (2018) propose a stream-based AL agent based on memory-augmented neural networks where an LSTM-based agent learns to decide whether to predict a class label or to query the oracle. Matching Networks (Bachman et al., 2017) extensions allow for pool-based AL. Fang et al. (2017) use Deep Q-Learning in a stream-based AL scenario for sentence segmentation. In contrast to them we consider batch-mode AL with acquisition sizes $\geq 1$, and work on a pool-instead of a stream-settings. While Bachman et al. (2017) propose a strategy to extend the RL-based approaches to a pool setting, they do still not work on batches. Instead, we allow batches of arbitrary acquisition sizes. Fan et al. (2018) propose a meta-learning approach that trains a student-teacher pair via RL. The teacher also optimizes *data teaching* by selecting labeled samples from a mini-batch that lets the student learn faster. In contrast, our method learns to selects samples from an unlabeled pool, i.e., in a missing target scenario. The analogy of teacher-student is related, however, the objective, method and available (meta-)data to learn a good teacher (policy) are different.

**Multi-armed Bandit (MAB).** Baram et al. (2004) treat the online selection of AL heuristics from an ensemble as the choice in a multi-armed bandit problem. `COMB` uses the known EXP4 algorithm to solve it, and ranks AL heuristics according to a semi-supervised maximum entropy criterion (Classification Entropy Maximization) over the samples in the pool. Building on this Hsu & Lin (2015) learn to select an AL strategy for an SVM-classifier, and use importance-weighted accuracy extension to EXP4 that better estimates each AL heuristics' performance improvement, as an unbiased estimator for the test accuracy. Furthermore, they reformulate the MAB setting so that the heuristics are the bandits and the algorithm selects the one with the largest performance improvement, in contrast to `COMB`'s formulation where unlabeled samples are the bandits. Chu & Lin (2016) extend Hsu & Lin (2015) to a setting where the selection of AL heuristics is done through a linear weighting, aggregating *experience* over multiple datasets. They adapt the semi-supervised reward scheme from Hsu & Lin (2015) to work with their deterministic queries. In our own work, we instead learn a unified AL policy instead of selecting from a set of available heuristics. This allows our policy to learn *interpolation* between batches of samples proposed by single heuristics and furthermore, to exploit the classifier's internal state, so that it is especially suited for deep learning models.

**Imitation Learning (IL).** Liu et al. (2018) propose a neural network that learns an AL strategy based on the classifier's loss on a validation set using Dataset Aggregation (DAGGER) (Ross et al., 2011). One of their key limitations is that only a single sample is labeled during every acquisition. As the DL model is trained from scratch after every acquisition this results in a very slow active learning process and expensive expert-time is requested less efficiently (Kirsch et al., 2019; Sener & Savarese, 2018). Hence, we extend this work for batch-mode AL using a *top-k*-like loss function, and select more samples to increase the suitability to deep learning and its efficiency (as we do not retrain after each sample). We also incorporate recent ideas (Ash et al., 2020) to extend the state and imitate multiple AL heuristics. This is computationally more efficient and leads to better results.

## 3 IALE: IMITATING AN ENSEMBLE OF ACTIVE LEARNERS

IALE learns an AL sampling strategy for similar tasks from *multiple experts* in a *pool-based* setting. We train a policy with data consisting of states (i.e., that includes an encoding of the labeled data samples) and best expert actions (i.e., samples selected for labeling) collected over the AL cycles. The policy is then used on a similar (but different) task. To see states that are unlikely to be produced by the experts, DAGGER (Ross et al., 2011) collects a large set of states and actions over AL iterations. The policy network is trained on all the previous states and actions after each iteration.

### 3.1 BACKGROUND

In pool-based AL we train a model $M$ on a dataset $\mathcal{D}$ by iteratively labeling data samples. Initially, $M$ is trained on a small amount of labeled data $\mathcal{D}_{lab}$ randomly sampled from the dataset. The rest of the data is considered as the unlabeled data pool $\mathcal{D}_{pool}$, i.e., $\mathcal{D} = \mathcal{D}_{lab} \cup \mathcal{D}_{pool}$. From that point onwards during the AL iterations a subset of $\mathcal{D}_{sel}$ is selected from $\mathcal{D}_{pool}$ by using an acquisition function $a(M, \mathcal{D}_{pool})$. The data is labeled and then removed from $\mathcal{D}_{pool}$ and added to $\mathcal{D}_{lab}$. The size of $\mathcal{D}_{sel}$ is based on the acquisition size $acq$ ($>1$ for batch-mode AL). The AL cycle continues until a labeling budget of $\mathcal{B}$ is reached. $M$ is retrained after each acquisition to evaluate the performance boost with respect to the increased labeled dataset only (and not the additional training time).

The acquisition function $a$ is a heuristic that uses the trained model $M$ to decide which of the data samples in $\mathcal{D}_{pool}$ are most informative. For deep AL popular heuristics include uncertainty-based `MC-Dropout` (Gal et al., 2017), query-by-committee-based `Ensembles` (Beluch et al., 2018), data diversity-based `CoreSet` (Sener & Savarese, 2018), gradient-based `BADGE` (Ash et al., 2020) and soft-max-based `Confidence`- or `Entropy`-sampling (Wang & Shang, 2014).

`MC-Dropout` uses a Monte-Carlo inference scheme based on a *dropout* layer to approximate the model's predictive uncertainty (Gal & Ghahramani, 2016). The heuristic (Gal et al., 2017) then uses these values to select the most uncertain samples. `Ensembles` (Beluch et al., 2018) model predictive uncertainty using a committee of $N$ classifiers initialized with different random seeds. However, while at inference time we need to run only $N$ forward-passes per sample (compared to `MC-Dropout` performing two dozen or more Monte-Carlo passes), the training of $N-1$ additional deep models can become prohibitively expensive in many use-cases. `CoreSet` (Sener & Savarese, 2018) aims to select diverse samples by solving the $k$-centers problem on the classifier's embeddings. This involves minimizing the distance between each of the unlabeled data samples to its nearest labeled samples. `BADGE` uses (pseudo labels of) the magnitudes of the gradients in a batch to select samples by uncertainty, and the gradient directions together with a k-means++ clustering to select samples by diversity. Soft-max-based heuristics (`Confidence`- and `Entropy`-sampling) use predictive uncertainty and are computationally lightweight at lower AL performance (Gal & Ghahramani, 2016; Ash et al., 2020) (`Confidence` selects the samples with the lowest class probability and `Entropy` the ones with with largest entropy of their probability distribution).

### 3.2 LEARNING MULTIPLE EXPERTS

Instead of using specific heuristics we propose to learn the acquisition function using a policy network. Once the policy is trained on a source dataset it can be applied to different target datasets. Figure 1 sketches the idea. The policy network $\pi$ is a Multi-Layer Perceptron (MLP) trained to predict the *usefulness* of labeling samples from the unlabeled data pool $\mathcal{D}_{pool}$ for training the model $M$, similar to an AL acquisition function. As input the policy network takes the current *state*, consisting

Figure 1: Training $\pi$ to imitate experts $\mathcal{E}$: (1) we pass samples from $D_{sub}$ and $D_{lab}$ through the current classifier $M$; (2) the embeddings and the predictions are input to $\pi$, whose output vector is compared with the target vector predicted by the best expert; (3) we calculate a loss and back-propagate the error through $\pi$; (4) we extend the labeled pool data $\mathcal{D}_{lab}$ by $\mathcal{D}_{sel}$ and retrain $M$.

of the model $M$'s embeddings of pool data next to other elements, extending similar work (Contardo et al., 2017; Konyushkova et al., 2017; Liu et al., 2018), see below. $\pi$ then outputs the *action* to be taken at that step. *Action* here refers to an AL acquisition, i.e., which of the unlabeled data samples should be labeled and added to the training data. $\pi$ learns the best *actions* from a set of experts $\mathcal{E}$ which predict the best actions for a given AL state. A subset of the pool dataset $\mathcal{D}_{sub}$ with size $n$ is used instead of the whole pool dataset at each active learning iteration for training the policy.

**States.** As $\pi$ uses the state information to make decisions, a state $s$ should be maximally compact but still unique, i.e., different *situations* should have different state encodings. Our state encoding uses two types of information: (1) model-dependent parameters (that describe the state from the perspective of the model $M$ and the already labeled samples $D_{lab}$), and (2) AL-cycle-dependent parameters (that describe the elements of the samples $\mathcal{D}_{sub}$ that we can choose to label). Together, these parameters form a minimal description of a current state.

We use the following parameters to describe the *model-dependent* aspects:

- The mean of the already labeled data samples $\mu\big(M_e(\mathcal{D}_{lab})\big)$: the embedding $M_e$ of a sample by $M$ is the output of the final layer (i.e., the layer before the soft-max layer in case of a classification model), see Figure 1. The size of this representation is independent of the (growing) size of $\mathcal{D}_{lab}$ and thus will not become a computational bottle-neck.

- The ground-truth empirical distribution of class labels

$$\vec{e}_{\mathcal{D}_{lab}} = (\frac{\sum_{y \in \mathcal{D}_{lab}} 1[y == 0]}{|\mathcal{D}_{lab}|}, ..., \frac{\sum_{y \in \mathcal{D}_{lab}} 1[y == i]}{|\mathcal{D}_{lab}|}),$$

  which is a normalized vector of length $i$, i.e., the number of classes, with percentage of occurrence per class using the labels of the already acquired data samples.

- $M$'s predicted empirical distribution of class labels for the labeled data $\vec{e}_{M(\mathcal{D}_{lab})}$ (i.e., a normalized vector as above but with predicted class labels instead of ground-truth).

The rationale for including both the ground truth and the predicted empirical distribution is to enable the policy to base its decisions on the model $M$'s prediction errors. In other words, when the model makes mistakes on already labeled samples for a class (that were part of the model's training) it likely needs more samples from those classes to correct the erroneous predictions.

The *AL-cycle-dependent* parameters describe the $n$ data samples in $\mathcal{D}_{sub}$ that we evaluate in the current iteration. For each data sample $x_i \in D_{sub}$ we calculate $M$'s embedding $M_e(x_i)$ in the same embedding space as the already labeled samples of our model-dependent parameters. We also predict each sample's label $M(x_i)$. Similarly, we capture the expected model change via per-sample gradient information $g(M_e(x_i))$ in the embedding space for $D_{sub}$, using the method proposed in (Ash et al., 2020). Hence, we consider the gradients of the loss at the embedding layer, given unlabeled samples with proxy labels, both as predictive of model uncertainty and expected model change. The proxy label $\hat{y}$ is the most likely prediction $M(x_i)$ (determined by an argmax operation in the softmax layer). The magnitude of the loss gradient at the embedding layer then describes both the model's uncertainty and its expected change.

This information enables the policy to learn to select samples (1) where the model is uncertain (i.e., where it predicts the wrong labels), (2) where the model might gain most information (i.e., the loss is high), and (3) to learn to select more diverse samples from less well represented classes using the label statistics. Hence, we describe a state $s$ as follows:

$$s := \left[ \mu(M_e(\mathcal{D}_{lab})), \vec{e}_{\mathcal{D}_{lab}}, \vec{e}_{M(\mathcal{D}_{lab})}, \begin{pmatrix} M_e(x_0) \\ .. \\ M_e(x_n) \end{pmatrix}, \begin{pmatrix} M(x_0) \\ .. \\ M(x_n) \end{pmatrix}, \begin{pmatrix} g(M_e(x_0)) \\ .. \\ g(M_e(x_n)) \end{pmatrix} \right] \qquad (1)$$

**Actions.** The *action* of the MLP is a *desirability score* $\rho_i$ for each unlabeled sample from $\mathcal{D}_{sub}$, i.e., $\rho_i := \pi(s_i)$. We choose the samples to be labeled based on this score, i.e., we choose the top-$k$ ranked values, where $k = acq$, resulting in a binary selection vector $\vec{v} = \text{top-k}(\rho_0, \cdots, \rho_n)$ with $\sum_{i=0}^{n} \vec{v}_i = acq$. The ground truth *actions* provided by the experts are binary vectors of length $k$, where a 1 at index $i$ means that $x_i$ should be selected for labeling (and indices of those sample that should not be labeled are 0). From here we can use a *binary cross entropy loss* to update $\pi$'s weights:

$$\mathcal{L}(\rho, \vec{t}) = \sum_{i=0}^{n} \vec{t}_i \log(\rho_i) - (1 - \vec{t}_i) \log(1 - \rho_i), \qquad (2)$$

where $\vec{t}$ is the target vector provided by the best expert (similar to a greedy multi-armed bandit approach (Hsu & Lin, 2015)). This brings $\pi$'s output closer to the suggestion of the best expert.

Our IL-based approach uses the experts to turn AL into a supervised learning problem, i.e., the action of the best expert becomes the label for the current state $s$. Our choice of AL heuristics for the set of experts $\mathcal{E}$ includes particular types but is arbitrarily extendable. Using `MC-Dropout`, `Ensemble`, `CoreSet`, `BADGE`, `Confidence` or `Entropy` allows us to only minimally modify the classifier model $M$. $\pi$ aims to learn certain derived properties from the *state*, such as model uncertainty or similarity. These measures are based on $M$'s predictions and embeddings.

Our hypothesis is that $\pi$ learns to imitate the best suitable heuristic for each phase of the AL cycle, i.e., starting with relying on one type of heuristics for selections of samples in the beginning and later switch to a *fine-tuning* using a different one (see also Section 4.3). This is in line with previous research that combines uncertainty- and density-based heuristics and that learns an adaptive combination framework that weights them over the training course (Li & Guo, 2013).

### 3.3 POLICY TRAINING

Our policy training builds on the intuition behind DAGGER, which is a well-known algorithm for IL that aims to train a policy by iteratively growing a dataset for supervised learning. The key idea is that the dataset includes the states that are likely to be visited over the course of solving a problem (in other words, those state and action encodings that would have been visited if we would follow a hard-coded AL strategy). To this end, it is common when using DAGGER to determine a policy's next state by either *following* the current policy or an available expert (Ross et al., 2011). We thus grow a list of state and action pairs, and randomly either choose expert or policy selections as the action.

**Algorithm 1:** Imitating Active Learner Ensembles

**Input:** data $\mathcal{D}$, labeled validation data $\mathcal{D}_{val}$, classifier $M$, budget $\mathcal{B}$, experts $\mathcal{E}$, acquisition size $acq$, subset size $n$, probability $p$, states $\mathcal{S}$, actions $\mathcal{A}$, random policy $\pi$ ($acq \geq 1$, $n = 100$).

2 **for** $e = 1 \dots episodes_{max}$ **do**
3    $\mathcal{D}_{lab}, \mathcal{D}_{pool} \leftarrow \text{split}(\mathcal{D})$
5    **while** $|\mathcal{D}_{lab}| < \mathcal{B}$ **do**
6       $M \leftarrow \text{initAndTrain}(M, \mathcal{D}_{lab})$
7       $\mathcal{D}_{sub} \leftarrow \text{sample}(\mathcal{D}_{pool}, n)$
8       $e^* \leftarrow \text{bestExpert}(\mathcal{E}, M, \mathcal{D}_{sub}, \mathcal{D}_{val})$
9       $\mathcal{D}_{sel} \leftarrow e^*.\text{SelectQuery}(M, \mathcal{D}_{sub}, acq)$
10       $\mathcal{S}, \mathcal{A} \leftarrow \text{toState}(\mathcal{D}_{sub}, \mathcal{D}_{lab}), \text{toAction}(\mathcal{D}_{sel})$
11       **if** $Rnd(0,1) \geq p$ **then**
12          // We may choose $\pi$'s selection
13          $\mathcal{D}_{sel} \leftarrow \pi.\text{SelectQuery}(M, \mathcal{D}_{sub}, acq)$
14       $\mathcal{D}_{lab} \leftarrow \mathcal{D}_{lab} \cup \mathcal{D}_{sel}$
15       $\mathcal{D}_{pool} \leftarrow \mathcal{D}_{pool} \setminus \mathcal{D}_{sel}$
16       Update policy using $\{\mathcal{S}, \mathcal{A}\}$

Each episode of the IL cycle lasts until the AL labeling budget is reached for $episodes_{max}$ iterations. We aggregate the *states* and *actions* over all episodes, and continually train the policy on the pairs.

Figure 2: Examples for the three datasets `MNIST`, `Fashion-MNIST`, and `Kuzushiji-MNIST`.

We use DAGGER to further randomize the exploration of $\mathcal{D}$. Instead of always following the best expert's advice, we randomly follow the policy's prediction, and thus enrich the possible states.

Our IL approach for training $\pi$ is given in Algorithm 1. At each AL cycle, we randomly sample a subset $\mathcal{D}_{sub}$ of $n = 100$ samples from the unlabeled pool $\mathcal{D}_{pool}$ (line 3). We find the best expert $e^*$ from a set of experts $\mathcal{E}$ (line 8) by extending the training dataset by the expert selections (from $\mathcal{D}_{sub}$) and train a classifier each. This means that each expert constructs one batch according to its heuristic, e.g., a batch composition could maximize model-change, and queries the oracle for labels. We choose the best expert by comparing the resulting classifiers' accuracies on the labeled validation dataset. We next set its acquisition as this iteration's chosen target and store *state* and *action* for the policy training (line 10). According to DAGGER we then flip a coin and depending on the probability $p$ (line 11) either use the policy or the best expert to increase $\mathcal{D}_{lab}$ for the next iteration (line 14). After each episode we retrain $\pi$ on the *state* and *action* pairs (line 16).

## 4 EXPERIMENTS

We first describe our experimental setup (Section 4.1). Next, we describe how we trained our policy (Section 4.2) and evaluate our approach by transferring it to test datasets, i.e., to `FMNIST` and `KMNIST` (Section 4.3). Finally, we end with a discussion of our ablation studies and the limitations of our approach (Section 4.4). The source code is publicly at `https://github.com/crispchris/iale` and can be used to reproduce our experimental results.

### 4.1 EXPERIMENTAL SETUP

**Datasets.** We use the image classification datasets `MNIST` (LeCun et al., 1998), `Fashion-MNIST` (FMNIST) (Xiao et al., 2017), and `Kuzushiji-MNIST` (KMNIST) (Clanuwat et al., 2018) for our evaluation. They all consist of $70,000$ grey-scale images ($28\times28$px) in total for 10 classes. `MNIST` contains the handwritten digits $0 - 9$, `FMNIST` contains images of clothing (i.e., bags, shoes, etc.), and `KMNIST` consists of Hiragana characters, see Figure 2.

To evaluate `IALE` we train a policy $\pi$ and run it (3 repetitions) on unseen datasets along with the baselines. The similarity between `FMNIST` and `MNIST` (that has previously been shown (Nalisnick et al., 2019)) and the difficulty of `FMNIST` (it has been shown to be a demanding dataset for AL methods (Hahn et al., 2019b)) make these datasets a perfect combination to evaluate `IALE`. Please find more results for transfering $\pi$'s AL strategy learned from `MNIST` to `CIFAR-10` in Appendix A.3.3.

**Architectures of classifier $M$.** We use the same model used to evaluate AL on `MNIST` data that has been used in previous research (Gal & Ghahramani, 2016). Our model has two convolutional layers, followed by a max pooling and dense layer. We add dropout layers after the convolution and dense layers and use ReLU activations. A softmax layer allows for classification. We also provide additional results for all the methods with a Resnet-18 (He et al., 2016) and with a two-layer MLP with a dense layer (256 neurons) followed by a soft-max layer (Ash et al., 2020) in Appendix A.3.2.

**Architecture of policy $\pi$.** Our policy model $\pi$ uses an MLP with three dense layers with 128 neurons each. The first two dense layers are followed by a ReLU activation layer, whereas the final layer has only one neuron and the output of this layer is passed onto a sigmoid function to constrain the outputs to the range $[0, 1]$ and to further process it into an aggregating top-k operation.

**Baselines.** For the evaluation of the performance of our AL method, we implemented different well-known AL approaches from literature: `Random Sampling`, `MC-Dropout`, `Ensemble`, `CoreSet`, `BADGE`, `Confidence-sampling`, `Entropy-sampling` and `ALIL` (which we adapted

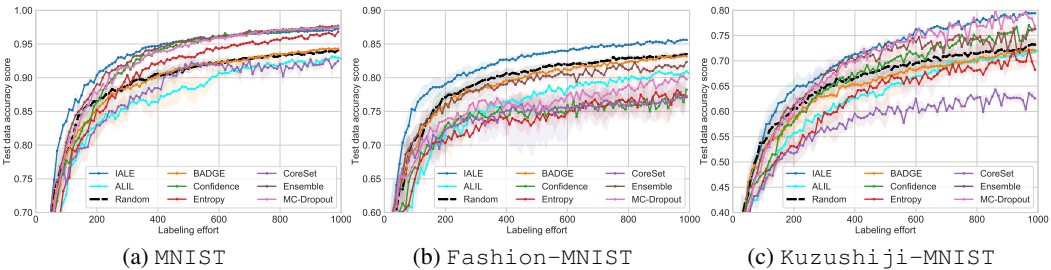

(a) MNIST          (b) Fashion-MNIST          (c) Kuzushiji-MNIST

Figure 3: Active learning performance of the trained policy (trained on MNIST), compared to the baseline approaches and ALIL (Liu et al., 2018), and applied to MNIST, FMNIST and KMNIST datasets.

to work on image classification tasks). For a more detailed description of the baselines please see Appendix A.1.1.

## 4.2 POLICY TRAINING

We use the MNIST dataset as our source dataset on which we train our policy for 100 episodes, with each episode containing data from an AL cycle. The initial amount of labeled training data is 20 samples (class-balanced). At each step of the active learning process, 10 samples are labeled and added to the training data until a labeling budget $\mathcal{B}$ of $1,000$ is reached. We use the AL heuristics MC-Dropout, Ensemble, CoreSet, BADGE, Confidence and Entropy as experts, and use $\mathcal{D}_{val}$ with 100 labeled samples to score the acquisitions of the experts. The pool dataset is sampled with $K = 100$ at each AL iteration. We choose $p = 0.5$ for DAGGER for means of comparison with the baselines (based on preliminary experiments (see Appendix A.2.1 on *Exploration-Exploitation*). We train the policy's MLP on the current and growing list of state and action pairs using the binary cross entropy loss from Equation 2 and use the Adam optimizer (Kingma & Ba, 2015) for 30 epochs with a learning rate of $1e - 3$, $\beta_1 = 0.9$, $\beta_2 = 0.999$, $\epsilon = 1e - 8$, and without any weight decay.

Figure 3a shows the results of our method in comparison to all the baseline approaches on the MNIST dataset, on which the policy was trained on. Our method consistently outperforms or is at least *en par* (towards the end, when enough representatives samples are labeled) with all the other methods. IALE performs better acquisitions than Ensemble and MC-Dropout for the important first half of the labeling budget, where it matters the most. Moreover, IALE is faster (see Appendix A.3.3). While MC-Dropout requires 20 forward passes to decide which samples it acquires, and Ensembles $N = 5$ forward passes, one for each model, our approach requires only 2 inferences for only $n = 100$ samples plus the labeled pool. Confidence-sampling performs similar as the two more complex methods, even though it uses only the simple soft-max probabilities. While Entropy beats random it is still not competitive. BADGE performs similar to random sampling, which is due to the small acquisition size of 10 (the better performance of BADGE was reported with much larger acquisition sizes of 100 to 10,000 in Ash et al. (2020) as its mix of uncertainty and diversity heuristic benefits from these). The same applies to CoreSet, however, here it performs worst on average over all experiments. This finding is in line with previous research (Sinha et al., 2019; Hahn et al., 2019a) and can be attributed to a weakness of the utilized $p$-norm distance metric regarding high-dimensional data, called the *distance concentration phenomenon*. The accuracy of ALIL on MNIST is similar to CoreSet, however, ALIL is designed to add only one sample to the training data at a time (no batch-mode).

**Construction of acquisition.** We compare IALE's chosen samples with the ones chosen by the baselines, see Figure 4. We show this overlap in relation to the baselines in percent, and plot the values over the 100 AL cycles using the MNIST dataset (more results can be found in Appendix A.2.2). We plot second-order polynomials, fit to the percentages (given as dots) over 100 acquisitions of size 10. $\pi$ mostly imitates uncertainty-based heuristics, i.e., soft-max heuristics and MC-Dropout, and the uncertainty-/diversity-heuristic BADGE (close behind). Interestingly, Ensemble is overlapping mostly at the beginning. CoreSet has the lowest overlap. Note that IALE's acquisitions are build

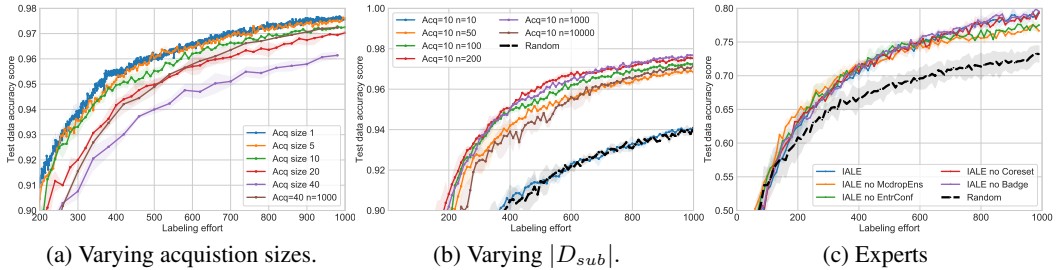

(a) Varying acquistion sizes.    (b) Varying $|D_{sub}|$.    (c) Experts

Figure 5: Ablation studies: IALE performance for different (a) $acq$ and (b) $n=|D_{sub}|$ (on `MNIST`); (c) IALE performance for different expert sets (on `KMNIST`).

from combinations of the heuristics (instead of single votes). The percentages do not sum up to 1 as the experts are independent and may also overlap with each other.

### 4.3 POLICY TRANSFER

We investigate how $\pi$ works on a different dataset than the one that it has been trained on. Hence, we train $\pi$ on the source dataset `MNIST` as in Section 4.2 and use it for the AL problem on `FMNIST` and `KMNIST`. For the AL we again use an initial class-balanced labeled training dataset of 20 samples and add 10 samples per AL acquisition cycle until we reach a labeling budget of 1,000 samples. All the baselines are evaluated along with our method for comparison.

Figures 3b and 3c show the performance of `IALE` along with the baselines on `FMNIST` and `KMNIST`. `IALE` consistently outperforms the baselines on both datasets. We can see that it learns a combined and improved policy

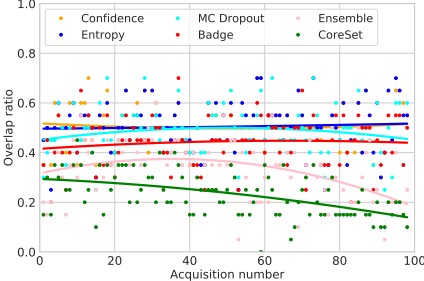

Figure 4: Overlap ratio.

that outperforms the individual experts consistently and (sometimes) even with large margins. On `FMNIST` `IALE` is the only method that actually beats `Random Sampling` (similar findings have previously been reported by Hahn et al. (2019b)). `IALE` is consistently $1-3\%$ better than `Random Sampling` on `FMNIST`, on the harder `KMNIST` dataset `IALE` is even $7-9\%$ ahead. The baselines give a mixed picture. `ALIL` does not achieve competitive performance on any task and actually never beats a random sampling strategy. We also see unstable performance for `MC-Dropout` and `Ensemble`, that generally perform similarly well. The simple soft-max heuristics `Entropy` and `Confidence` fail on `FMNIST`. `CoreSet` lags far behind, especially on `KMNIST`. `BADGE` always performs like random sampling, due to the aforementioned problematic acquisition size. Please find additional experiments in Appendix A.3 including a wider range of classifier models (MLP, CNN, and Resnet-18) and datasets (`CIFAR-10`).

### 4.4 ABLATION STUDIES

**Hyperparameters.** Two important parameters are the acquisition size $acq$ and the size of $\mathcal{D}_{sub}$. Figures 5a and 5b show results for $acq$ of 1 to 40 and size of $\mathcal{D}_{sub}$ between $n=10$ and $n=10,000$. As expected, `IALE` performs best at $acq=1$ and worst at $acq=40$, if $n$ is unchanged, because $n$ limits the available choices, e.g., bad samples have to be chosen. Increasing $n$ to $1,000$ alleviates this issue. However, there is an upper limit to the size of $\mathcal{D}_{sub}$ after which performance deteriorates again, see Figure 5b. This could be because random sub-sampling actually simplifies the selection of diverse, uncertain samples. The lower limit becomes apparent again when $n$ is smaller than 10 times $acq$, with $n = acq$ essentially being a random sampling. From our observations, $n$ should be $10-100$ times $acq$. From the small differences within this value range, it is suggested that our method is suitable for larger acquisition sizes for batch-mode AL, as its performance is not affected much.

**Varying experts.** To investigate the influence of experts, we leave out some types of experts: We categorize them into 4 groups, i.e., uncertainty (`McdropEns`), soft-max uncertainty (`EntrConf`), diversity (`Coreset`) and hybrid (`Badge`), and leave one subset out. We fully train each method on `MNIST` with $\mathcal{B} = 1,000$ and an acquisition size of 10, and present the results of the evaluation on `KMNIST` in Figure 5c (more results including the ablation of state elements can be found in Appendixes A.4.4 and A.4.3). We see that most combinations perform well compared to the baselines. However, leaving out uncertainty or soft-max uncertainty experts can decrease performance. Even though training time is longer with `MC-Dropout`, the gain in performance can be worth it. In contrast, the soft-max uncertainty based heuristics are computationally cheap and yield good policies.

## 5 CONCLUSION

We proposed a novel imitation learning approach for active learning. Our method learns to imitate the behavior of different active learning heuristics, such as uncertainty-, diversity-, model change- and query-by-committee-based heuristics, on one initial dataset and model, and transfers the obtained knowledge to work on other (types of) datasets and models (that share an embedding space). Our policy network is a simple MLP that learns to imitate the experts based on embeddings of the dataset samples. Our experiments on well-known datasets show that we outperform the state of the art consistently (despite being a batch-mode AL approach). An ablation study and analysis of the influence of certain hyper-parameters also shows the limitations of our approach. Future work investigates the relationship between acquisition sizes and sub-pools, and an analysis on how the *state* (embedding) enables $\pi$ to transfer its active learning strategy, potentially leading to the use of artificial data for finding optimal AL strategies suitable for specific active learning scenarios.

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

# A APPENDIX

In this section we provide an extension of the experiments section (Section 4) and feature additional results that support a more complete evaluation of `IALE`. We adhere to the same section structure.

## A.1 EXPERIMENTAL SETUP

### A.1.1 BASELINES

In the following is a short explanation of the baselines and experts that we used in our experiments:

1. `Random Sampling` randomly samples data points from the unlabeled pool.
2. `MC-Dropout` (Gal et al., 2017) approximates the sample uncertainty of the model by repeatedly computing inferences of the sample, i.e., 20 times, with dropout enabled in the classification model
3. `Ensemble` (Beluch et al., 2018) trains an ensemble of 5 classifiers with different weight initializations. The uncertainty of the samples is quantified by the disagreement between the model predictions.
4. `CoreSet` (Sener & Savarese, 2018) solves the $k$-center problem using the pool-embeddings of the last dense layer (128 neurons) before the softmax output to pick samples for labeling.
5. `BADGE` (Ash et al., 2020) uses the gradient of the loss (given pseudo labels), both its magnitude and direction, for k-means++ clustering, to select uncertain and diverse samples from a batch.
6. `Confidence`-sampling (Wang & Shang, 2014) selects samples with the lowest class probability of the soft-max predictions.
7. `Entropy`-sampling (Wang & Shang, 2014) calculates the soft-max class probabilities' entropy and then selects samples with the largest entropy, i.e., where the model is least certain.
8. `ALIL` (Liu et al., 2018): we modify ALIL's implementation (that is initially intended for NLP tasks) to work on image classification task. Due to the high runtime costs of running ALIL (as the acquisition size is 1), we perform the training of `ALIL` for 20 episodes. We trained the ALIL policy network with a labeling budget $\mathcal{B}$ of $1,000$ and an up-scaled policy network comparable to that of our method along with a similar $M$ as we use to evaluate the other AL approaches. We left the coin-toss parameter at $0.5$, and the $k$ parameter for sequential selections from a random subset of $\mathcal{D}_{pool}$ at 10.

We use the variation ratio metric (Gal et al., 2017) to quantify and select the data samples for labeling from the uncertainty obtained from `MC-Dropout` and `Ensemble` heuristics. The variation ratio metric is given by its Bayesian definition (Gal et al., 2017) for a data sample $x \in \mathcal{D}_{\text{pool}}$ in Equation 3 and for an ensemble expert (Beluch et al., 2018) in Equation 4:

$$\text{variation-ratio}(x) = 1 - max_y p(y|x, D) \tag{3}$$

$$= 1 - \frac{m}{N}, \tag{4}$$

where $m$ is number of occurrences of the mode and $N$ is the number of forward passes or number of models in the ensemble.

## A.2 POLICY TRAINING

### A.2.1 EXPLORATION-EXPLOITATION IN DAGGER

DAGGER uses a hyper-parameter $p$ that determines how likely $\pi$ predicts the next action, and thereby setting the next state, instead of using the best expert from $\mathcal{E}$. In this preliminary study we compare the influence it has to either fix $p$ to $0.5$ or to use an exponential decay parameterized by the number of the current episode $epi$: $1 - 0.9^{epi}$. We train the policy on `MNIST` for 100 episodes with a labeling budget of $1,000$ and an acquisition size of 10 (as before). Our result is that the *fixed* policy

outperforms the *exponential* one by a small margin for the transfer of the policy to another dataset than the trained one, which is in line with previous findings (Liu et al., 2018).

A balanced (i.e., fixed) ratio does not emphasize one over the other, whereas an exponentially decay quickly relies on the policy for selecting new states of the dataset, and thus it trains on too few optimal states over the AL cycle.

### A.2.2 OVERLAP RATIOS

We additionally show `FMNIST` and `KMNIST` overlap ratios in Fig. 6. The policy chooses about half of the samples differently from any single baseline. The overlap is lower on `FMNIST`, where our method is the only one that beats random sampling. `CoreSet` overlaps least often and performs worst.

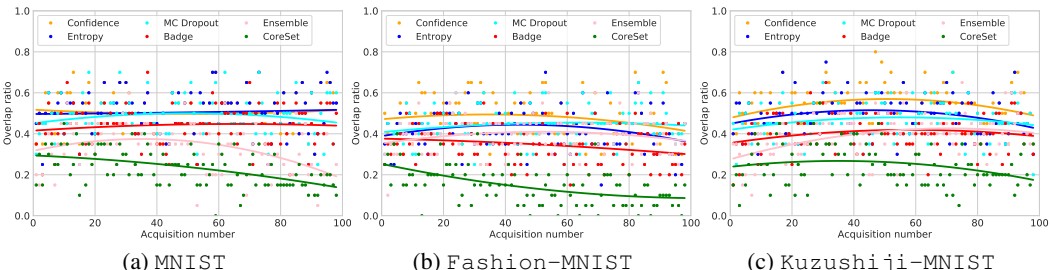

(a) `MNIST`  (b) `Fashion-MNIST`  (c) `Kuzushiji-MNIST`

Figure 6: The overlap plots for all datasets `MNIST`, `FMNIST` and `KMNIST` datasets.

### A.3 POLICY TRANSFER

In this section we provide further studies on how our method performs in regard to applying it to unseen scenarios. These include that we (1) train the policy on other source datasets than `MNIST`, that we (2) apply the policy onto other classifiers than the one it was trained on, and that we (3) use different datasets and classifier models in training and application of the policy.

We aim to show that $\pi$ learns a (relatively) task-agnostic AL strategy, that works in inference and outperforms the baselines, as long as the *state* can be expressed similarly to the one that has been used during training.

### A.3.1 VARYING SOURCE DATASETS

Our policy does not coincidentally perform well due to a specific *source* dataset that is most suitable. In Figure 7, we show two additional policies that were trained on `FMNIST` or `KMNIST` (with unchanged hyper-parameter settings). We see comparable performance that indicates that `IALE` actually *learns to actively learn* and not just remembers the source datasets as it makes not difference which datasets are chosen as the source and the target dataset.

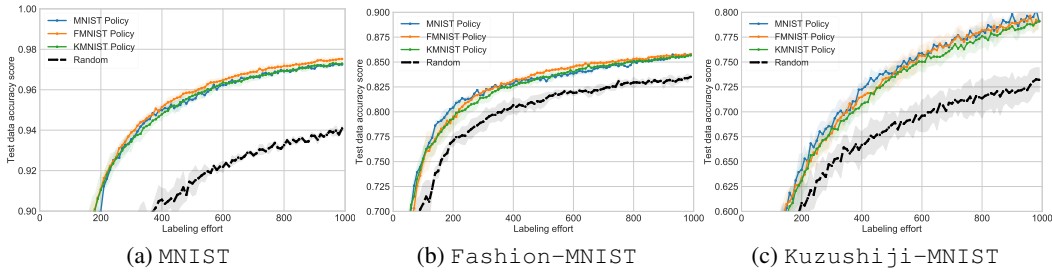

(a) `MNIST`  (b) `Fashion-MNIST`  (c) `Kuzushiji-MNIST`

Figure 7: We show the performance of three policies trained on different datasets in comparison with `Random` (on `MNIST`, `FMNIST` and `KMNIST`).

### A.3.2 CLASSIFIER ARCHITECTURES.

Our method is not bound to a specific classifier architecture. To show this, we evaluate our approach with MLP and ResNet-18 classifiers. We use the same hyper-parameters as with the CNN and only switch out the underlying classifier. All results for the experiments are given in Figure 8. We see the robustness of $\pi$ over fundamentally different classifier architectures (2 to 18 layers). The deviations for ResNet-18 are very large due to the very deep architecture and the modest amount of training data. We use median filtering in Figures 8g, 8h, and 8i.

These experiments show that $\pi$ can learn AL strategies for both very small and very deep architectures and still outperform baselines. Even though the strongest baselines, i.e., `CoreSet` and `MC Dropout`, come close to our method in accuracy, they are less versatile and require more computational resources, that is especially noticeable on deeper architectures.

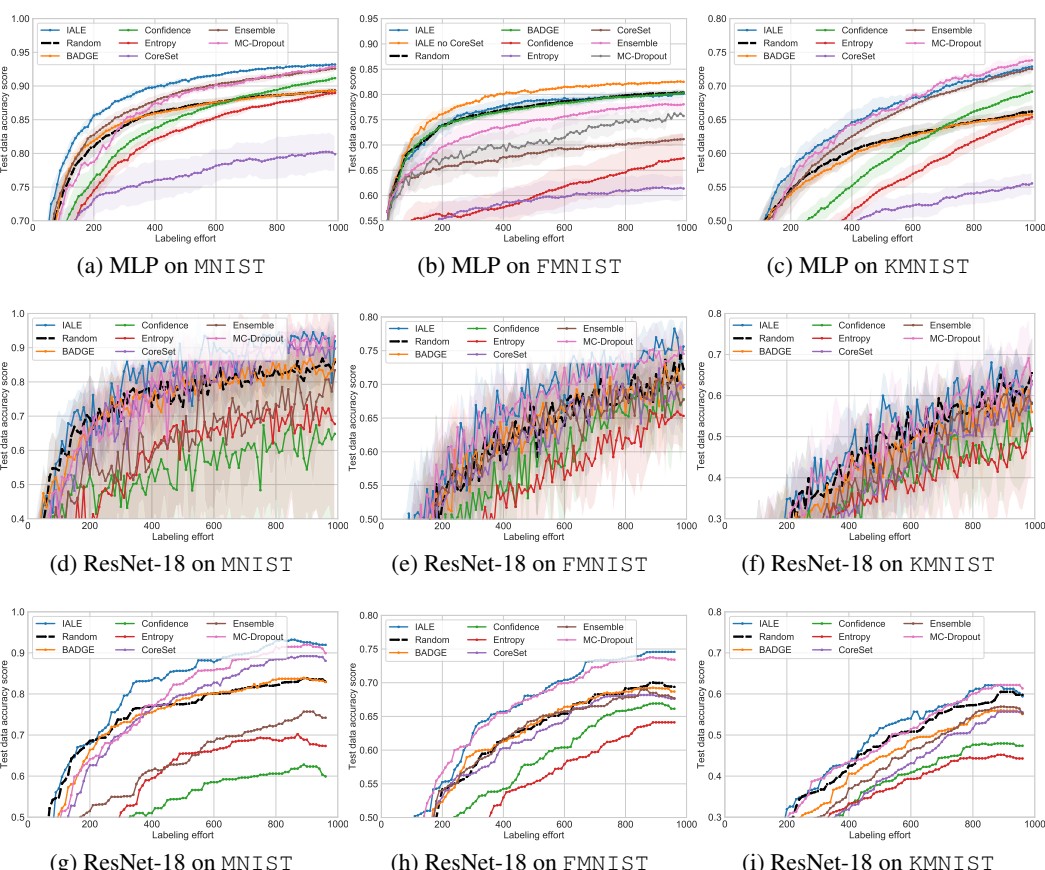

Figure 8: MLP and ResNet-18 classifiers, data averaged or median filtered. Active learning performance of the trained policy in comparison with the baseline approaches on `MNIST`, `FMNIST` and `KMNIST` datasets.

### A.3.3 GENERALIZATION

To show that $\pi$ learns active learning independent from task and classifier, we conduct experiments where we mix both the source datasets *and* the classifiers, as only the *state* retains the same formulation. Since these states are the same for CNN and Resnet-18 classifiers (independant of the datasets) we can transfer a policy trained using a Resnet-18 classifier to a CNN classifier and vice versa.

We report the results for applying $\pi$ (trained on Resnet-18 and `MNIST`) to a CNN and all the datasets in Figure 9. `IALE` is always performing at the top. These results convincingly show that our method

learns a *model- and task-agnostic* active learning strategy that transfers knowledge between datasets and even between classifier architectures.

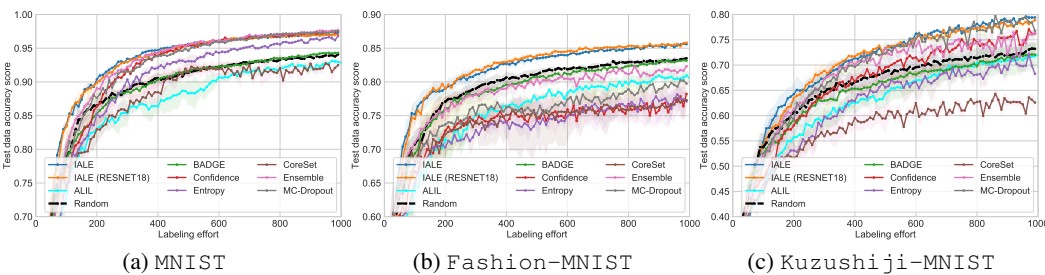

Figure 9: Applying a policy trained using a Resnet-18 classifier (trained on `MNIST`) to a CNN-based classifier (on `MNIST`, `FMNIST` and `KMNIST`).

We conclude the transfer studies by applying pre-trained $\pi$ to a Resnet-18 classifier on the `CIFAR-10` (Krizhevsky, 2009) (color) image classification dataset. Each image in `CIFAR-10` is $32 \times 32$ pixels and has 3 color channels. The transfer of $\pi$ is possible as the classifier's `state` is both invariant to the models and independent from the number of color channels of the image dataset and the input size. For our experiments we use two different $\pi$: $\pi_1$ was trained using Resnet-18 and `MNIST` (`IALE Resnet`) and $\pi_2$ was trained using CNN and `MNIST` (`IALE CNN`). We train the classifiers with an acquisition size of 10 until the labeling budget of 10.000 is reached.

The results are presented in Figure 10. As the acquisition size of 10 results in noisy curves we report the raw learning curves (Figure 10a) and median filtered learning curves (Figure 10b). We report the interesting segment of the learning curve in more detail (filtered) in Figures 10c. The results generally show the feasibility of transferring $\pi$ to both different classifiers and datasets. `IALE` is *en par* or better than `Random`, and the other baselines are either en par or worse than `Random` (some of them considerably).

Moreover, besides `IALE`'s better accuracy compared to all other methods, its run time is highly competitive. Per experiment iteration (with 10.000 samples, acquired over 998 steps of size 10, each step including a complete re-training of Resnet-18 for 100 epochs), run times for `IALE` are 10:17:31 (h:m:s) on a high performance GPU (NVidia V100) (9:45:12 for `Random` sampling), compared to 49:15:23 for `Ensembles`, 14:23:18 for `MC-Dropout` and 11:58:47 for `BADGE`. Only `Conf` (10:12:01) and `Entropy` (10:05:21) are quicker (however, they both also perform worse than `Random`).

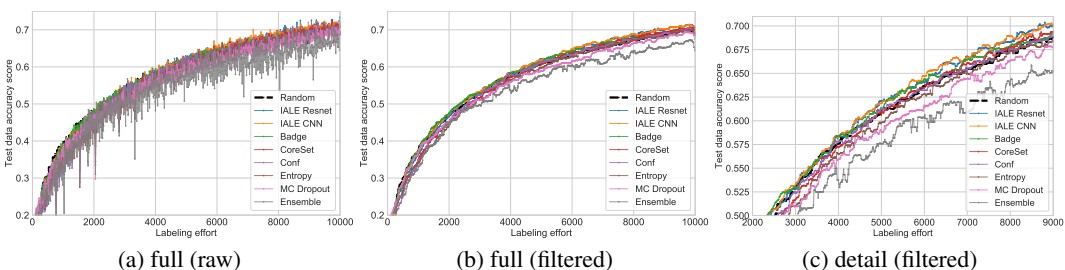

Figure 10: Full and enlarged segments of learning curve: Applying $\pi$ trained on a CNN (IALE CNN) or Resnet-18 (IALE Resnet), trained on `MNIST`, to a Resnet-18 classifier on `CIFAR10`.

While more experiments are certainly required to further emphasize these initial claims of generalizability to more diverse types of datasets, these findings are already very promising.

### A.4 ABLATION STUDIES

#### A.4.1 HYPERPARAMETERS.

We report fine-granular steps of acquisition sizes in Figure 11a, with values between 1 and 10, plus 20 and 40, for $|D_{sub}|$ of 100. Overall, a clear difference is not visible below 10 samples. For enhanced readability, we show a magnified section of the varied acquisition sizes and $|D_{sub}|$ in Fig. 11b, that clearly shows the benefits of tuning $|D_{sub}|$ to a suitable value for the acquisition size.

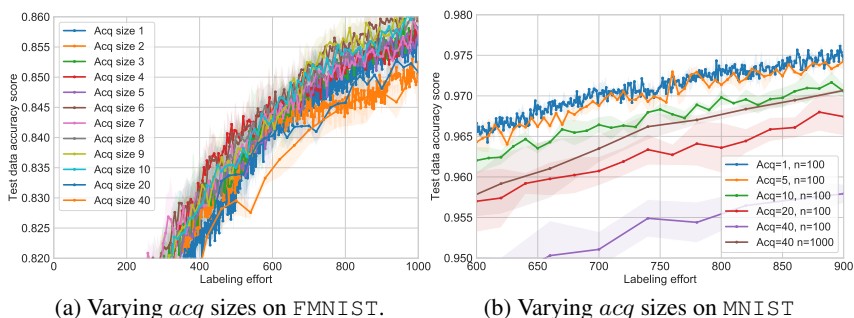

(a) Varying *acq* sizes on `FMNIST`.  (b) Varying *acq* sizes on `MNIST`

Figure 11: Evaluating the acquisition sizes from 1 to 10 on `FMNIST`, and varying different pool sizes on `MNIST`.

**Acquisition sizes for baselines:** We additionally compare the baseline active learning methods with our approach, as these exhibit different performance at different acquisition sizes, see Fig. 12. We have included comparisons with acquisition sizes of either 1 or 100 (1 or 3 repetitions). For our method, for an acquisition size of 1 we chose $|D_{sub}| = 100$ and for acquisition size of 100 we chose $|D_{sub}| = 2,000$. While the results show that `IALE` outperforms the baselines they also highlight the large effect that the acquisition size has on some of the baseline methods. For instance, `CoreSet` constructs better set covers with larger batches, and `BADGE` increases its accuracy by constructing a representative sampling as well. At the same time the uncertainty-based methods, apart from `Entropy`, remain unaffected.

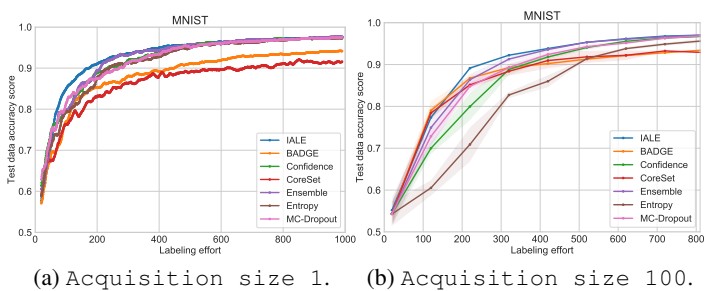

(a) Acquisition size 1.  (b) Acquisition size 100.

Figure 12: Some experts are more suitable to other acquisition sizes (evaluated on `MNIST`).

#### A.4.2 VARYING EXPERTS

We present more results for variations of sets of experts in Figure 13. We leave out some types of experts, and train the policies with the unchanged hyper-parameters and the CNN classifier. The results for all three datasets show that the generally high performance of `IALE` holds for the leave-one-out sets of experts, with the full set of experts being consistently among the best performing policies.

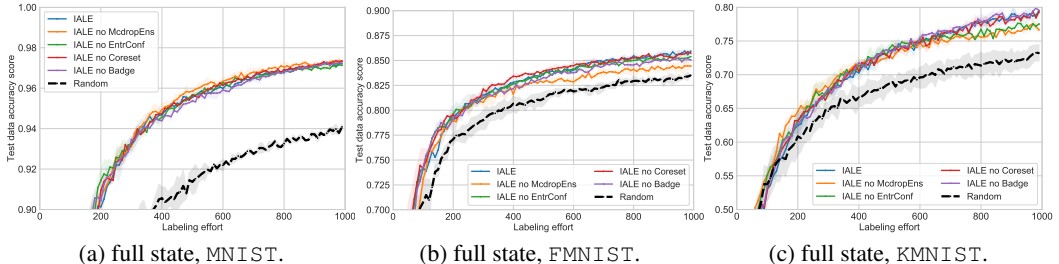

(a) full state, `MNIST`.     (b) full state, `FMNIST`.     (c) full state, `KMNIST`.

Figure 13: The active learning performance for each (leave-one-out) set of experts.

### A.4.3 VARYING STATE ELEMENTS

Next, we study the state more closely. For unlabeled samples, the state contains two types of representations for predictive uncertainty: the statistics on predicted labels $M(x_n)$ and the gradient representations $g(M_e(x_n))$. In this study, we focus on leaving out one or the other. To get the full picture, we again train sets of experts for reduced states.

In Figure 14 we see that dropping gradients generally decreases performance (bottom row), while dropping predicted labels $M(x_n)$ affects performance very little (top row). However, the influence of different sets of experts is more important. We cannot see that a particular set of states and experts generally outperforms others consistently (while the negative effect of leaving out $g(M_e(x_n))$ is consistently visible). Overall, we find that using as many experts as available, combined with a full state both performs well and works reliably. Even though training a policy this way does not guarantee the best performance, it always performs among with the group of best policies.

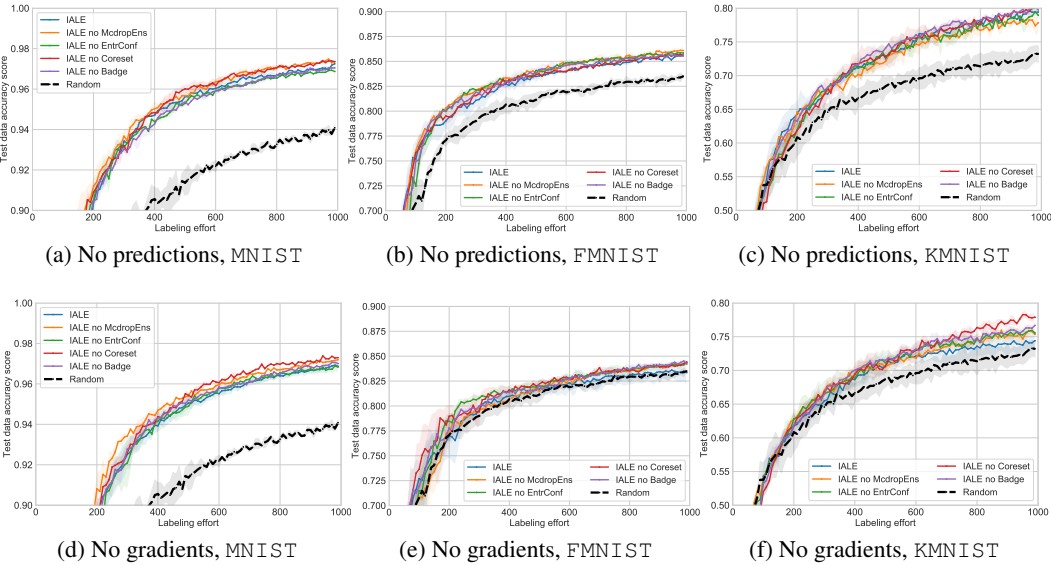

(a) No predictions, `MNIST`     (b) No predictions, `FMNIST`     (c) No predictions, `KMNIST`

(d) No gradients, `MNIST`     (e) No gradients, `FMNIST`     (f) No gradients, `KMNIST`

Figure 14: For partial state (without predictions, without gradients), we plot active learning performance for each (leave-one-out) set of experts.

### A.4.4 ADDITIONAL DATASETS

We additionally run experiment on harder datasets, i.e., `SVHN` and `CIFAR-100` in Fig. 15.

**SVHN:** We train a Resnet18 on SVHN with an acquisition size of 1,000 and a labeling budget of 16,000. We initially label 1,000 samples and evaluate over 5 repetitions. We show both the average results with variance in Fig. 15a and the smoothed averages for improved visibility in Fig. 15b. While the results exhibit some variance, we can clearly see that `IALE` performs best (and is the only AL methods that is consistently able to beat a random sampling.

**CIFAR-100:** Next, we evaluate datasets with a larger number of classes by generalizing the composition of the *state* (removing *prediction* and *empirical class distribution*). We train two policies, one with a CNN classifier on `MNIST` and the second with a Resnet18 classifier on `CIFAR-10`. Then, we apply the resulting policies in the following experiment. We train a Resnet18 on the very hard `CIFAR-100` dataset, with an acquisition size of 1,000 and a labeling budget of 20000, and initially label 1,000 samples. We evaluate over 5 repetitions. Fewer samples or less powerful network architectures tend to fail to converge in our experiments. The results in Fig. 15c show that both policies, i.e., transferred from CNN/MNIST and from Restnet18/CIFAR10 with an arbitrary number of classes, perform best among all heuristics, and beat random sampling.

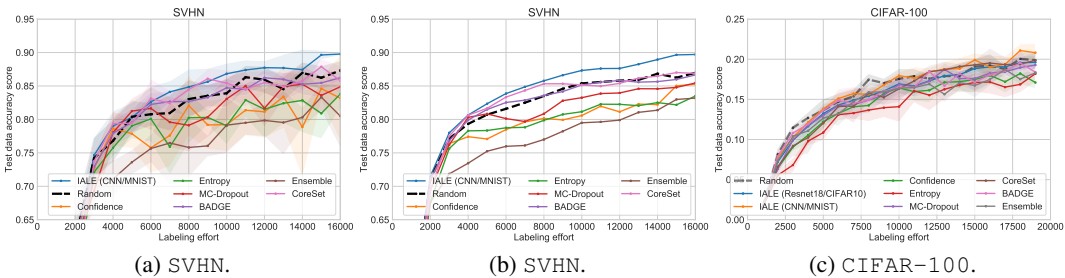

(a) SVHN.  (b) SVHN.  (c) CIFAR-100.

Figure 15: Increasingly difficult datasets: SVHN (complex structure, requires more samples, average and smoothed plots) and CIFAR-100 (100 classes).

