# OpenReview forum: "IALE: Imitating Active Learner Ensembles"
_ICLR.cc/2021/Conference — Reject_

### Official Review · AnonReviewer4 · 2020-10-28
**good idea for using imitation learning to combine the advantages of multiple active learners, while the approach and the experimental evaluation are not sufficiently sound**

**Rating:** 4
**Confidence:** 4

**Review:**

#### Paper summary:

In this work, an imitation learning (AL) approach is proposed to imitate multiple active learning algorithms, in order to take their advantages to learn a better active learning algorithm. The main idea is to treat the active learning algorithms as experts and utilize the DAGGER algorithm for imitation learning. The proposed approach is evaluated on MNIST, fashion-MNIST, and Kuzushiji-MNIST, showing that the learned active learner outperforms baseline active learners, meanwhile is transferrable to other datasets.

#### Advantages:

- The idea of using imitation learning is interesting.

- The survey of existing active learning approaches is quite thorough.

#### Disadvantages:

- The idea of using DAGGER to combine multiple active learners was explored in Liu et. al. https://www.aclweb.org/anthology/P18-1174. The technical novelty is thus limited.

- The experiments are not sufficient for the following perspectives:

  - The performance gain is somehow not very significant, the datasets are restricted to images with the same number of classes.

  - The policy utilizes the features of training instances as part of the states. While I am in doubt of why such features can be transferrable to new datasets?

  - There are no comparisons to the work of Liu et. al., which also utilizes DAGGER to imitate active learners.

Overall evaluation:

I think the paper proposes a reasonable solution, while the contributions and novelty are somehow below the standard of ICLR.

Additional:

- The methodology for the proposed approach is similar to the "learning to teach" approach https://openreview.net/pdf?id=HJewuJWCZ, which learns a teaching policy for selecting the smallest set of labeled instances to accelerate learning. Adding some discussions to their work can be useful.

- The "test set" in algorithm 1 should be "validation set".

#### Update after rebuttal:
Thanks for the detailed feedback. While I still find the technical novelty is limited comparing to [Liu et. al.]. For this reason, my score remains unchanged.

---

> ### Author Response · Authors · 2020-11-18
> **Author Feedback**
>
> We would like to thank the reviewer for the fruitful comments and the feedback. We addressed them in the submission accordingly.
>
> We would like to refer the reviewer on the comments we make on the differentiation to the work of Liu et al. in the general section above (as all reviewers have pointed that out). We also made changes to the submission to clear things up. Please comment on this if there are still unclear aspects. We also added comments on experiments, which was also requested by all of the reviewers (above).
>
> On the two remaining comments:
> 1. Additional RW : "Learning to Teach" --  We have extended the discussion on related work section with a discussion on "Learning to Teach".
> 2. Why are features of training instances transferable? -- We believe that the accumulated state/action-pairs capture transferable features similarly to how transfer-learning enables the re-use of image features in image classification, without re-learning convolution kernels from scratch.

---

### Official Review · AnonReviewer1 · 2020-10-28
**The proposed framework makes sense, but exact contributions need to be clarified.**

**Rating:** 6
**Confidence:** 3

**Review:**

The paper proposes a batch-mode active learning approach via imitation learning to derive a sample selection strategy for active learning. The proposed method makes use of a pool of existing heuristics, learning a sample selection strategy that rely on different heuristics in the pool depending on the different stage of the active learning process. The use of imitation learning makes sense, as it helps to alleviate the high sample complexity problem of deploying a reinforcement learning strategy. Furthermore, a batch-mode active learning strategy is desirable compared to the previous work (Liu et al. 2018) that uses a single sample. Experimental results on MNIST,  FMNIST, and KMNIST demonstrates the utility of the proposed method. I have the following concerns:

1. While I think the idea of framing the active learning problem as an imitation learning problem has merits, it is unclear to me how the proposed framework is different from that of (Liu et al. 2018). The authors point out that (Liu et al. 2018) achieves single-sample mode active learning but the proposed framework achieves batch-mode active learning. However, the way that batch-mode is achieved in this paper seems trivial by computing and using the top-k instances according to certain metrics.

2. It is also unclear how the pool of experts used in the paper generalizes beyond the setting in (Liu et al. 2018), and why this might be desirable. Given this concern and the previous one, it would be great if the authors can clarify the relationships between the two works so as to better highlight the specific contributions made in this work.

3. In terms of empirical validation, is it possible to also compare the proposed method with (Liu et al. 2018)? Furthermore, the authors demonstrate the utility of the proposed method in three related datasets (MNIST,  FMNIST, and KMNIST). I am not sure if such empirical validation is sufficient. How about other datasets that contain digits such as SVHN? Is it possible to imitate on CIFAR 10 and transfer on CIFAR 100? How about NLP tasks that are reported in (Liu et al. 2018)? On the other hand, I think the ablation studies provided by the authors do help to demonstrate how different components of the framework will play and influence the eventual outcome.

Miscellaneous:
1. There is a missing comma in the last sentence of the paragraph right before Section 3.1?

---

> ### Author Response · Authors · 2020-11-18
> **Author Feedback**
>
> We would like to thank the reviewer for his/her fruitful comments and the constructive feedback. We will run additional experiments and add them to the submission accordingly.
>
> Related to comments 1 and 2, we lined out a more detailed differentiation to the work of Liu et al. in the general comments section above (as this issue was raised among all reviewers) and also made changes to the submission. Please feel free to comment again if things remain unclear. As the questions on experiments were also shared by the other reviewers, we comment on these in the general comments section as well.
>
> We evaluated IALE on the requested datasets (CIFAR-100 and SVHN) and initial results already seem promising as IALE still outperforms all the other approaches, see the added Appendix A.4.4 at the end of the updated submission. We will run more trials and update the submission accordingly. However, a comparison against NLP tasks such as those reported in Liu et al. take too long and we are not able to evaluate our method on NLP datasets on this short notice. However, we believe that IALE is also transferable to this domain and still maintains its performance advantage as we do not see any reason why this should not be the case.

---

### Official Review · AnonReviewer2 · 2020-10-29
**Solving active learning by learning to imitate the heuristics**

**Rating:** 5
**Confidence:** 4

**Review:**

This paper proposes an approach to learning active learning policies that rely on imitating the behaviour of known heuristics. The resulting learnt policies can successfully transfer between the related datasets.

Strong points:
- This paper focuses on the important problem and attempts to provide a simple solution that leverages the prior knowledge about the problem (existing heuristic solutions) while being data-driven.
- The experimental studies are quite informative and detailed and provide analysis of the behaviour of the proposed solution.
- The proposed method outperformed the presented baselines in most cases.
- The experiments with policy transfer between different datasets seem very interesting and focus on the realistic and important setting.
- The experiments with ablations by excluding various experts are informative and insightful.

Weak points:
- The novelty of the proposed method is not very strong: The main idea is to select the best behaviour among several heuristics, that was explored before (Hsu&Lin, 2015, 1, 2, 3), the learning mechanism relies on imitation learning, that was also explored in previous works for AL (Liu et al, 2018), and the implementation in terms of state parametrization seem to be similar to previous works too (Contardo et al, 2017, Konyushkova et al., 2017, Liu et al, 2018).
- This paper puts an emphasis on the batch nature of the proposed method (for example, when contrasting against the related work). However, I did not understand how the policy training is particularly targeted to the batch setting.
- The nature of the proposed method is the most similar to the algorithms that select a particular heuristic to use at each step of the training. While the authors mention some of such algorithms (Hsu&Lin, 2015), I believe a few others works deserve some attention [1, 2, 3]. Importantly, there is no comparison to the methods from this group.
- The authors claim that the imitation learning is done by the DAGGER method. However, as far as I understand, DAGGER makes an assumption that an expert can provide an optimal action from any state as supervision. If I understand the method correctly, such an assumption cannot be satisfied in the setting of the paper. To my mind, it sounds that the method is closer to supervised learning with epsilon-greedy data collection.
- Most experiments are performed on rather low dimensional data in MNIST, FMINST and KMNIST. The transfer to more complex datasets such as CIFAR is shown in the appendix, but the gains by the proposed method seem to be marginal.


I am leaning towards the rejection of this paper in the light of limited novelty of the presented method and limited gains especially on more complex realistic datasets.

Questions:

- Section 4.4 contains a curious and counter-intuitive observation: "there is an upper limit to the size of D_{such} after which performance degrades again". However, the authors did not provide any explanation of this phenomena. Why is this happening?
- Could the authors elaborate a bit more on the connection between the proposed learning mechanism and DAGGER?
- I think the most significant part of the method is the ways how the training data is collected with supervision of the experts. While this procedure is covered in section 3.3, I think it deserves a bit more attention. For example, I am still not sure if and how *batches* of data are selected by heuristics strategies.
- Why do the authors say that the method is closer to the baselines "towards the end obviously"? Is the most of the data labelled at that point?
- The learning curves in many experiments seem to be quite noisy. How many experiments are averaged?
- How do the baselines compare to the proposed method with varying batch size (as some methods are particularly targeted to batch size=1 or large batch size>100)?

Additional comments:

- The language in the paper often uses unusual word ordering and constructions that makes it difficult to read. For example, "as ideally we label", "besides the above mentioned", "do still not work on batches".
- A lot of experimental results are presented by training and testing the policy on the same dataset. As this is a rather unrealistic setting and serves mostly as sanity check, less space in the paper could be dedicated to it.


[1] Online choice of active learning algorithms. Y. Baram, R. El-Yaniv, and K. Luz. JMLR, 2004.
[2] Can active learning experience be transferred? H.-M. Chu and H.-T. Lin. ICDM, 2016.
[3] Active learning by learning. W.-N. Hsu and H.-T. Lin. In American Association for Artificial Intelligence Conference, 2015

===Post rebuttal===

I have read the authors response and would like to thank the authors for the clarifications. I am still inclined to keep my original score.

---

> ### Author Response · Authors · 2020-11-18
> **Author Feedback**
>
> Thank you for the thorough review and the constructive feedback. The comments are very helpful for us and we take them all into account to modify the submission accordingly. We think that the submission improves a lot. There has been a confusion related to the differentiation to previous work presented by Liu et al. As all the reviewers raised this issue we decided to address this in the general discussion section above (there we also added comments on experiments, which was also requested by all of the reviewers). Please feel free to further comment on this if things still remain unclear.
>
> There were some additional comments on the novelty concerns regarding two other streams of research:
> - Related work that deals with selecting the best heuristic from an ensemble: Thank you for providing additional references (however, [2] was already included). We have extended the discussion of [2] with the additionally provided related work. Related work [1,2,3] in the multi-armed bandit (MAB) context estimates the benefits of choosing one AL heuristic from an ensemble over another via a test accuracy approximation. They use Classification Entropy Maximization or (modified) Importance-Weighted Accuracy. In contrast, we learn a unified policy that performs selections faster than a set of in-parallel executed heuristics, that learns to interpolate between AL heuristics (see Fig. 4) and that exploits the classifiers' internal state, and thus is especially suitable for deep learning models.
>
> - Related work that deals with the parameterization of the state: The state we propose extends existing work (Contardo et al., 2017, Konyushkova et al., 2017, Liu et al., 2018)), especially for deep models, and our initial submission shows the clear performance benefits (see Appendix 4.3 Fig. 13): Our proposed state composition includes approximations of the classifier's learning progress of the dataset, i.e., the uncertainty of different clusters of data. The state measures data distribution and expected model-change: We use the classifier's gradients at an embedding layer in order to measure the uncertainty of unlabeled data and its diversity, given pseudo-labels (compare Ash et al. 2020).
>
>
> Further questions:
> - Why do the authors say that the method is closer to the baselines ”towards the end obviously”? Is the most of the data labelled at that point? -- Towards the end enough representative samples are labeled. We updated the text accordingly.
> - How does policy training target the batch setting? Batch data selection in Sec. 3.3  -- The IALE policy learns to select diverse, uncertain batches by imitating AL heuristics that select batches. We have clarified that in Sec. 3.3.
> - DAGGER's optimal action assumption -- We agree with the reviewer on DAGGER's optimal action constraint, and reformulate Sec. 3.3 accordingly. We build on the intuition behind DAgger, and iteratively build a dataset that we use for supervised training. This is in line with similar work in various applications in which the requirement for an optimal supervisor is relaxed as well [4,5,6].
> - Large $D_{sub}$ degenerates performance -- We have clarified this observation in Sec. 4.4: "[the decrease in performance with large $D_{sub}$] could be because random sub-sampling actually simplifies the selection of diverse, uncertain samples", i.e. selection-bias is implicitly reduced by random sub-sampling.
> - How many repetitions per experiment? -- We added information on this to the submission.
>
>
> [1] H.-M. Chu and H.-T. Lin.  Can active learning experience be transferred? In ICDM, 2016.
>
> [2] W.-N. Hsu and H.-T. Lin.  Active learning by learning.  In American Association for Artificial Intelligence Conference, 2015.
>
> [3] R. El-Yaniv Y. Baram and K. Luz.  Online choice of active learning algorithms.
>
>
> [4] Steephane  Ross et al.: Learning monocular reactive uav control in cluttered natural environments.In2013 IEEE international conference on robotics and automation, 2013.
>
> [5] Ming Liu et al.: Learning how to actively learn: A deep imitation learning approach. In Proceedings of the 56th AnnualMeeting of the Association for Computational Linguistics, 2018.
>
> [6] Osbert  Bastani et al.: Verifiable  reinforcement learning via policy extraction. In Advances in neural information processing systems, pages 2494–2504, 2018.

---

### Author Response · Authors · 2020-11-18
**General Rebuttal Information**

We want to thank the reviewers for their constructive feedback. We are encouraged that the reviewers find the problem important (R2) and our approach of leveraging prior knowledge while being data-driven interesting (R1, R2, R4). The progress compared to related work in respect to batch-mode AL is desirable (R1), the method alleviates issues of related methods that use RL (R1) and its performance is a strong-point (R2), all while our survey of existing active learning approaches is quite thorough (R4). The reviewers found our experiments informative and detailed (R1, R2), specifically the analysis of components within the ablation studies (R1, R2) and the policy transfer (R2). Here, we address the reviewers' (shared) main points on (1) novelty over Liu et al. and (2) experiments. We answer fine-grained aspects of the particular reviews separately. We also made an update to the submission and addressed the issues raised by the reviewers. To simplify the discussion we made all changes to the submission in red color.

(1) Novelty over Liu et al. First, our approach proposes several novelties over ALIL: we (i) select diverse, uncertain batches per acquisition, (ii) learn faster (CPU time) involving a *diverse set* of AL heuristics, and (iii) use a very effective state representation, and (iv) transfer to tasks with arbitrary number of classes. We prove a considerable performance difference compared to ALIL in Fig. 3 (original submission). In our updated submission we clarified this naming convention in the figure's description to avoid confusion (as we actually did compare to ALIL already). Second, ALIL only imitates a single learner. Its imitated algorithmic expert is very slow and selects only one sample: it first randomly samples K=10 samples from the unlabeled pool (K>10 does not increase the performance of ALIL, see Liu et al.), retrains the classifier on each sample individually, and then selects the sample with the highest accuracy. This pseudo-random sub-sampling of the pool is neither suitable for large pools nor batches (diverse, uncertain), while our diverse set of AL heuristics samples from the whole pool. Furthermore, Liu et al. retrain the classifier 10 times, once for each single selected sample. This does not scale well. Instead, we train the classifier for each AL heuristic, which scales considerably better.

(2) Experiments. All the reviewers (R1, R2, R4) request more experimental results and insights (more experiments on SVHN, Cifar10 and Cifar100, and NLP (R1); varying batch-sizes with baselines (R2); more experiments with more classes (R4). We did our best and present new results in this short amount of time. As of now the number of repetitions is limited due to the required compute power for the AL evaluations. We include our preliminary additional results for SVHN and CIFAR-100 in (the additional) Appendix A.4.4 (last page), and also compare IALE to baselines for variations of the acquisition size in Appendix A.4.1. Upon acceptance we will move the final graphs (including averages over several repetitions) to the main part of the paper.

---

### Decision · Program_Chairs · 2021-01-07
**Final Decision**

**Decision:**

Reject

**Comment:**

In the discussion, all reviewers acknowledge the novelty of this paper, such as learning from a wide range of AL heuristics, and the ability to transfer the to tasks with arbitrary number of classes. They also think that the additional experiments provided by the authors improve the paper's empirical validity.

However, a major issue raised by the reviewers is that the novelty (especially when compared with Liu et al) may not be enough for ICLR this time. One research direction (implicitly suggested by Reviewer 2) was to learn active learning strategies that go beyond selecting top-k scoring examples to explicitly account for batch diversity. We encourage the authors to address this in the next version.